# Predicting the Prognostic Value of *POLI* Expression in Different Cancers via a Machine Learning Approach

**DOI:** 10.3390/ijms23158571

**Published:** 2022-08-02

**Authors:** Xuan Xu, Majid Jaberi-Douraki, Nicholas A. Wallace

**Affiliations:** 1Division of Biology, Kansas State University, Manhattan, KS 66506, USA; xuanxu@ksu.edu; 21DATA Consortium, Kansas State University Olathe, Olathe, KS 66061, USA; 3Department of Mathematics, Kansas State University, Manhattan, KS 66506, USA

**Keywords:** polymerase iota, cancer survival, machine learning, gene association, gene regulatory network

## Abstract

Translesion synthesis (TLS) is a cell signaling pathway that facilitates the tolerance of replication stress. Increased TLS activity, the particularly elevated expression of TLS polymerases, has been linked to resistance to cancer chemotherapeutics and significantly altered patient outcomes. Building upon current knowledge, we found that the expression of one of these TLS polymerases (*POLI*) is associated with significant differences in cervical and pancreatic cancer survival. These data led us to hypothesize that *POLI* expression is associated with cancer survival more broadly. However, when cancers were grouped cancer type, *POLI* expression did not have a significant prognostic value. We presented a binary cancer random forest classifier using 396 genes that influence the prognostic characteristics of *POLI* in cervical and pancreatic cancer selected via graphical least absolute shrinkage and selection operator. The classifier was then used to cluster patients with bladder, breast, colorectal, head and neck, liver, lung, ovary, melanoma, stomach, and uterus cancer when high *POLI* expression was associated with worsened survival (Group I) or with improved survival (Group II). This approach allowed us to identify cancers where *POLI* expression is a significant prognostic factor for survival (*p* = 0.028 in Group I and *p* = 0.0059 in Group II). Multiple independent validation approaches, including the gene ontology enrichment analysis and visualization tool and network visualization support the classification scheme. The functions of the selected genes involving mitochondrial translational elongation, Wnt signaling pathway, and tumor necrosis factor-mediated signaling pathway support their association with TLS and replication stress. Our multidisciplinary approach provides a novel way of identifying tumors where increased TLS polymerase expression is associated with significant differences in cancer survival.

## 1. Introduction

Genotoxic chemotherapeutic agents (e.g., cisplatin) are commonly used to treat multiple different types of tumors. These drugs typically kill cancer cells by causing DNA lesions that lead to replication stress. This affords some level of specificity as, in general, transformed cells are more likely to be replicating than other cells in the body. Genotoxic agents are generally effective treatment options, but resistance remains a significant barrier to success [1,2,3]. There has been a sustained effort to identify the molecular mechanisms by which resistance can be acquired so that targeted therapies can be designed for these drug-resistant cancers. This information can also be used to identify markers of tumors that will be resistant to standard interventions, allowing alternative approaches to be used.

The translesion synthesis (TLS) pathway has recently been identified as a mediator of resistance to genotoxic chemotherapies [4,5,6,7,8]. TLS allows replication forks to bypass DNA lesions caused by drugs, such as cisplatin. This prevents their collapse and the resulting toxicity that allows cisplatin to kill replicating cells. Mechanistically, TLS accomplishes this bypass by promoting the exchange of high-fidelity replicative polymerases for an error-prone TLS polymerase (e.g., *POLH*, *REV3L*, *POLI*) [9,10,11]. Whereas replicative polymerases cannot synthesize DNA using damaged DNA as a template, TLS polymerases can incorporate an untemplated base, allowing them to move beyond a DNA lesion [12]. TLS polymerase abundance appears to be rate-limiting for the pathway as exogenous TLS polymerase expression results in a more efficient DNA lesion bypass [13].

Typically, tumors with elevated TLS polymerase expression are significantly less responsive to genotoxic therapies, resulting in worse prognoses for people with these tumors [1,14]. However, this is not universally true as in some tumor types increased TLS polymerase expression is associated with improved prognosis. Cervical cancers (CESC) are an example of a tumor type where increased expression of a TLS polymerase (*REV1*, *POLH*, or *REV3L*) is associated with reduced survival, while pancreatic cancers (PAAD) are an example of the opposite [13,15,16]. This implies that the tumor cell environment, most likely the transcriptome, dictates the prognostic value of TLS polymerase expression for cancer survival.

If whether TLS polymerase expression acts as a positive or negative prognostic factor is dictated by changes in the transcriptome of tumor cells, then identifying the gene (s) most responsible for driving these differences is important. Traditional molecular biology approaches manipulations can help find these genes, but it has limited scalability and is further hampered by the lack of validated reagents to detect most proteins. Machine learning algorithms (e.g., decision tree, neural network (NN)) are not limited by either of these restraints and have been used to identify other cancer prognostic factors at much lower costs than in vitro screening would have required [15,16,17,18]. There are advantages and disadvantages to each machine learning algorithm. For example, an NN approach based on multi-layer perceptrons includes more complexity but produces less interpretability [15,19]. Unlike the black box nature of NN, decision trees, such as random forest (RF) [20] and gradient boosting (GB) [21], embrace features of simplicity and the “easy-to-learn” nature of a tree-structure algorithm [17].

In this manuscript, we probe transcriptomic data from the cancer genome atlas (TCGA) to identify genes associated with *POLI* expression being a positive/negative prognostic factor. This allowed us to build a classification system that successfully predicted a group of tumors where *POLI* expression would be positively associated with survival and a group of tumors where the relationship would be the opposite. These relationships were independent of tumor type. We also determined the extent to which the genes used to categorize tumors were enriched in cell processes. To achieve these goals, we employed a variety of computational tools, including a statistical learning approach for gene selection, a machine learning method for a supervised classifier, and data-drive approaches for gene network association to eventually investigate the relationship among genes, cellular processes, and cancer progression. Previous studies have shown that the quantitative measurement of gene correlations can be associated with cellular functions [22,23]. This was our rationale for using the expression data of genes correlating with *POLI* expression as the input for a cancer classifier [22]. Further, because decision tree algorithms have been used to identify cancer biomarkers [22,24,25,26], we used RF in our analysis over GB and NN.

## 2. Results

### 2.1. Prognostic POLI Expression Signatures

To confirm whether *POLI* was associated with both improved and worse outcomes varied by tumor type, we performed Kaplan–Meier (K–M) survival analysis on CESC and PAAD. *POLI* was a significant prognostic factor in both of these tumor types (Figure 1). However, we found that the increase in *POLI* expression was not always linked to a decrease in survival (Figure 1b). As a result, survival data of 11 other cancer types, including bladder (BLCA), breast (BRCA), colorectal (COADREAD), head and neck (HNSC), liver (LIHC), lung (LUAD and LUSC), ovarian (OV), melanoma (SKCM), stomach (STAD), and uterine (UCEC), were integrated and then standardized to determine the extent that *POLI* expression correlated with survival. We observed that *POLI* expression did not correlate with survival in these tumor types when combined (Figure 1c). The K–M analysis of individual cancer types also showed no prognostic value (Appendix A). These data show that the ability of *POLI* to act as a prognostic factor varies by tumor type.

### 2.2. POLI-Associated Genes

To investigate the regulation of *POLI* expression, associated genes were selected as the candidate variables according to the working hypothesis. We computed the pairwise Pearson correlation coefficient between *POLI* and the remaining genes for CESC and PAAD, respectively (Table 1). To determine the extent that these results were dependent on the manner in which correlations were determined, we repeated this analysis using a non-parametric method (Spearman’s rank correlation). This analysis produced a similar range of correlation values demonstrating that the analysis was largely independent of the correlation method employed (Appendix A). The top positively and negatively correlated genes were collected for CESE and PAAD. Generally, a stronger association was observed in PAAD patients compared to CESC.

To refine and identify the list of most correlated genes to *POLI* expression, the graphical least absolute shrinkage and selection operator (GLASSO) was applied to the merged 1000 genes from the top positive and negative columns (Table 1) to encourage further sparsity. Due to the noise or weak signals in gene data, the tuning parameters of GLASSO were adjusted to select 200 genes from each cancer type, CESC, and PAAD, respectively. These 200 genes out of 1000 genes from either CESC or PAAD were considered as the most positively/negatively correlated genes potentially upregulating or downregulating *POLI* expression. The 2 sets of the 2 genes from the 2 cancers were merged forming 1 set of 396 unique genes, with only 4 genes overlapping in the 2 cancer types. This group of genes indicated the features of gene expression patterns influencing *POLI* in CESC and PAAD.

### 2.3. Random Forest Classifier for CESC and PAAD

After gene selection and dimension reduction, genes integrated from CESC and PAAD correlating to *POLI* were considered as the dependent variables for a supervised machine learning process. We performed a binary classification task via random forest classifier (RFC), based on selected genes with the contrary prognostic value of *POLI* in CESC and PAAD. Patients in CESC and PAAD were under-sampled to gain an unbiased classification of two cancer types. RFC reached 100% accuracy to differentiate CESC and PAAD on the 30% testing data using 396 gene variables.

For new testing patients with other cancer types, RFC calculated the similarity between Group 1, “CESC-like” containing potential signal of increase in *POLI* expression worsening survival, and Group 2, “PAAD-like” containing potential signal of increase *POLI* expression improving survival. Our RFC were extended to segregate people with 11 other cancers, based on whether their gene expression was more similar to “CESC” or “PAAD” (Table 2). An amount of 54% of patients were classified as group 1. In total, the classification yielded a balanced result for 11 cancers. However, individual cancer types showed varied preferences toward two predicted groups.

### 2.4. Identifying Tumors Where POLI Expression Will Correlate with Survival

After the classification of patients from other cancers, we asked whether the prognostic value of *POLI* existed in patients clustered in Group 1 and Group 2, respectively. Based on our previous results in CESC and PAAD. The distinct relationship between *POLI* expression and survival was expected to show in the two groups. The backward selection was performed to select cancer types that the prognosis of *POLI* expression would stand out from Group 1 and Group 2, respectively (Table 2). Patients with BRCA, STAD, and UCEC were classified into Group 1 and Group 2 regarding *POLI* as a prognostic factor. HNSC, LIHC, and SKCM from Group 1 when merged with BRCA, STAD and UCEC showed the pattern as we demonstrated in CESC that upregulated *POLI* expression worsened the survival. BLAC, COADREAD, LUAD, LUSC, and OV when integrated with BRCA, STAD and UCEC showed the pattern of *POLI* expression in PAAD, which was the opposite phenomenon (Figure 2). The gene expression signature and RFC parameters provided sound support for patient classification concerning *POLI* and TLS. To determine the extent that this classification was influenced by the method used to determine correlations, we repeated this analysis using genes selected via Spearman’s rank correlation (Appendix A). The results in selected Group 1 and Group 2 showed the same patterns in Figure 2, indicating the robust nature of our results. Further, we performed a comparative analysis of different machine learning approaches (RF, GB, and NN). These data demonstrated that RFC outperform the other classifiers, as BG and NN approaches were less capable of clustering patients with the same prognostic value of *POLI* (Appendix A).

### 2.5. Analysis of Gene Association Pattern

To determine and extract genes highly correlated, we performed statistical learning and visualization approaches to reduce the dimension and reveal gene patterns and associations integrating the analytical procedures from previous computational studies [27,28]. Gene importance in RFC was obtained first to get the top 100 genes, which accounted for 89% of input for our classification effort (Appendix A). Then, genes were re-processed via GLASSO to reduce the size to 50 highly associated genes using the gene–gene correlation matrix of CESC and PAAD, respectively. We found that 22 genes existed in both cancer types. Fifty genes formed one large and six small clusters in CESC, and two large clusters in PAAD indicate gene–gene interaction conditioning on *POLI* expression and potential factors regulating *POLI* expression (Figure 3).

In CESC, AURKAIP1 was associated with 25 genes, forming an isolated cluster consisting of 36 genes. Other independent clusters contained two to four highly associated genes. In PAAD, CPLX2 was associated with 12 genes in one of the large clusters containing 30 genes in total. *MRPL4*, *PSMB6*, *RSPH9*, *SURF2*, *CLPP*, *DNAH7*, *MRPL12*, *ISOC2*, *C21orf70*, *BRMS1*, *EDF1*, *SF3B5*, *NAP1L3*, *NSUN5*, *DPM2*, *SALL2*, *SPATA4*, *RPS15*, *MGMT*, *NDUFB11*, *SV2A*, and *RPL28* existed in CESC and PAAD regarding the top 50 highly associated genes. AURKAIP1 and CLPX2 existed in a much sparser gene network with a reduction to 25 vertices in CESC and PAAD (Appendix A).

We were also able to convert the circus plot to a heatmap layout, which showed gene-gene clusters and partitions in a comprehensive way. Vertical gene names were hidden when they were out of the top 50 genes in RFC. Almost half of the genes in CESC (28/50) and PAAD (24/50) were the top 50 genes evaluated by RFC.

### 2.6. Gene Ontology Enrichment Analysis

We next considered the known functional relationship among the genes selected from RFC using the gene ontology enrichment analysis and visualization tool (GOrilla) [29]. These ranked biological processes based on the extent that genes selected by RFC were enriched. The top five biological processes are summarized in Table 3, which also includes the ranking (by RFC) of the individual genes found to be enriched in each biological process. We also indicated whether these 100 genes were involved in the 50 most associated genes in CESC, PAAD, or not. This analysis identified enrichments in pathways linked with mitochondrial activities, Wnt signaling, and tumor necrosis factor-mediated (TNF-mediated) signaling pathways.

## 3. Discussion

Here, we describe our efforts to identify a subset of genes capable of classifying tumors into two groups; one where increased *POLI* expression will correlate with improved odds of survival (pancreatic-like or Group II) and the other with the opposite relationship between *POLI* expression and patient outcome (cervical-like or Group I). To achieve this goal, we applied multiple computational approaches combining supervised and unsupervised machine/statistical learning methods to address the classification, feature selection, and network analysis. This supports our hypothesis that the prognostic value of *POLI* expression is determined by the transcriptome of an individual tumor. Changes in the prognostic value of gene expression could be influenced by mutations in the gene that resulted in a different interactome. To determine the likelihood that the results reported here were driven by *POLI* mutations among tumor types, we determined the frequency of *POLI* mutations in each TCGA database used for this analysis. Specifically, *POLI* mutations were found in only 3.0% of these tumors overall and there were only small variations in *POLI* mutation frequency among the tumor types analyzed in this study (1.4–5.0%). Our analysis does not support the hypothesis that *POLI* mutations are a robust determinant of the prognostic value of *POLI* (not shown).

The insights provided here could serve as the basis for improving the use of biomarkers to guide cancer therapies. Currently, individual markers (or a panel of markers) are used to guide patient care. For example, platinum-based therapies (e.g., cisplatin) might not be used in someone with a high expression of *POLI*. However, our data suggest that elevated *POLI* expression does not always manifest in resistance to platinum-based drugs. Further, our work implies that it is possible to use transcriptomic data to predict when *POLI* will or will not be a useful indicator of resistance to therapy.

Computational cancer studies usually focus on a specific cancer type. We proposed a methodical way to merge and normalize patients with different cancers. Normally, clinical data, i.e., age, gender, diagnoses, and smoking history were considered as the input in Cox proportional hazard regression model for risk classification [30]. In this study, the prognostic value of *POLI* was not demonstrated in cancers other than CESC and PAAD before the binary classification. To avoid arbitrary diagnosis using data from a large population of specific cancers, varied gene signatures of patients should be taken into account. Our approach incorporated unsupervised learning on gene selection and supervised learning on classifying patients to build the classifier that can discriminate between a positive and negative prognosis value of *POLI* expression. This approach connected apparent randomness and sophisticated gene correlation. Previous studies have shown the success of connecting a subset of genes to survival [31]. 

There are gaps not addressed in our work. Most obviously, we do not directly consider resistance to platinum-based drugs in our analysis, using survival data as an indirect metric of responsiveness to therapy. Although platinum-based drugs are widely used, patient outcomes are influenced by a myriad of factors and our analysis does not address this nuance [32]. Further, the classification scheme described here was not able to segregate several types of cancers based on their ability to predict this prognostic value of *POLI* expression. Thus, there are several areas where our work can be refined through future efforts.

Another area where continued effort is warranted is in determining the biological mechanisms that dictate whether *POLI* is a positive or negative prognostic factor. This understanding could allow specific therapeutic targets to be identified with the potential to improve outcomes for all tumors with/without increased *POLI* expression. We began addressing the biological significance of the genes in our classifier in this study using GO enrichment analysis of the top 100 genes identified by RFC. This analysis linked the genes to several biological processes of known significance in tumorigenesis. For instance, there was a significant enrichment in genes involved in TNFα and WNT signaling. These pathways act as tumor suppressors. We also found enrichment in biological functions associated with cell metabolism (i.e., mitochondrial elongation and termination). *AURKAIP1* has been identified as a valuable feature It has been shown that *AURKAIP1* promotes Aurora-A, an oncogene, the overexpression of which attributes to aneuploidy and could lead to cancer potentially [33]. Unlike *AURKAIP1*, which was negatively associated with *POLI* expression in CESC and PAAD, *CPLX2* only showed a significantly strong association with *POLI* expression in PAAD. Our survival analysis of patients in Group 2 included two lung cancer datasets, LUAD and LUSC (Figure 2b). It has been reported that *CPLX2* could be a reasonable biomarker in high-grade lung cancer [34]. As the one highly associated with other genes in PAAD, for those patients clustered in Group 2, *CPLX2* regulated *POLI* expression might mediate patient survival. It is notable that of many of the genes and biological processes that could have been included in our classification scheme, we found such an enrichment for those closely linked with tumorigenesis.

Finally, in this study, we have compared the feasibility of decision-tree based models (i.e., RF and GB) with NN for cancer patient classification. Results corroborate the robustness and flexibility of RFC over the other two algorithms for the given dataset and settings (Appendix A). Due to the capability of RF in handling a large set of gene features with a lot of background noise simultaneously without overfitting and massive hyperparameter tuning, RF outperforms NN. This suggests that RF is the better machine learning approach when identifying transcriptomic changes that influence the prognostic value of a gene of interest.

## 4. Materials and Methods

### 4.1. Data curation

RNA sequence data of 11 cancer types and clinical data were downloaded and integrate to 1DATA databank from the Broad GDAD Firehouse (http://gdac.broadinstitute.org, accessed on 20 June 2022), including CESC (309 patients) [35], PAAD (183 patients) [36], BLCA (427 patients) [37], BRCA (1212 patients) [38], COADREAD (433 patients) [39], HNSC (566 patients) [40], LIHC (423 patients) [41], LUAD (576 patients) [42,43], LUSC (552 patients) [42,43], OV (307 patients) [44], SKCM (473 patients) [45], STAD (450 patients) [46], and UCEC (201 patients) [47]. The number of fragments per kilobase of exon per million reads of *POLI* expression data was downloaded from the Human Protein Atlas (https://www.proteinatlas.org/ (accessed on 21 November 2021)). The different steps of data processing, filtering, and feature selection from curation to survival analysis via machine and statistical learning were described in the flowchart (Figure 4).

### 4.2. Genes Associated with POLI in CESC and PAAD

Genes used to build the classifier were selected from CESC and PAAD separately. A total of 1000 genes, which consisted of the top 500 positively associated with *POLI* and the top 500 negatively associated were combined for CESC and PAAD, respectively. GLASSO was applied to introduce the sparsity to the inverse covariance matrix for gene–gene correlation and select the highly associated 200 genes from 1000 merged genes [48]. A total of 200 genes from CESC and another 200 genes from PAAD were joined as the final gene list as the classifier features.

RNA sequence data were log-transformed to have the standardization for each gene and each cancer type, separately. The final size of gene variables was 396 after merging 200 genes from CESC and PAAD.

### 4.3. Random Forest Classifier

The opposite relationship between *POLI* and survival in CESC and PAAD was the basis that genes associated with *POLI* in two cancers would perform classification and can be extended to calculate the similarity in other cancer types as well. RFC was built using 396 genes selected previously to have the model and parameter. For this step, the under-sampling method was applied to eliminate the impact due to more patients in CESC that the classifier would not predict more testing data to CESC. An amount of 70% of CESC and PAAD were kept as the training data for the classifier. The other 30% of CESC and PAAD data were used to validate the performance of the classifier. The data of patients in 11 other cancers were then differentiated through a model based on CESC and PAAD.

### 4.4. Survival Analysis

For cancer data integration, survival time was divided by the max length of date in individual cancer type and multiplied by 1000. The K–M curve was used for the analysis of study groups with over-expression or under-expression *POLI* and survival time [49].

### 4.5. Gene Function Validation

GOrilla was used to identify enriched gene ontology terms regarding 396 genes built for the cancer classifier [29]. The top 100 genes from RFC via the feature importance method were tuned via GLASSO down to the 50 most associated genes focusing on the core regulating *POLI* in CESC and PAAD, respectively.

### 4.6. Visualization

The circus layout was set to show the association between 50 genes for pathway validation. The reverse Cuthill–Mckee (RCM) reordering method was employed to permute sparse matrices into a band matrix that associated genes were reordered toward the diagonal [50]. The connection between genes was encouraged to have less crossing in order to unmask the core circus plot of gene clustering.

### 4.7. Softwares

We performed K–M survival and gene network analysis using R version 4.1 and Python 3.9. RCM was implemented in MATLAB R2019b (version 9.7; MathWorks Inc., Natick, MA, USA; RRID: SCR_001622). GLASSO, K–M plots, and circos plots were generated using R package *huge, survival,* and *edgebundleR.* Heatmaps were generated Python module *Seaborn*.

## Figures and Tables

**Figure 1 ijms-23-08571-f001:**
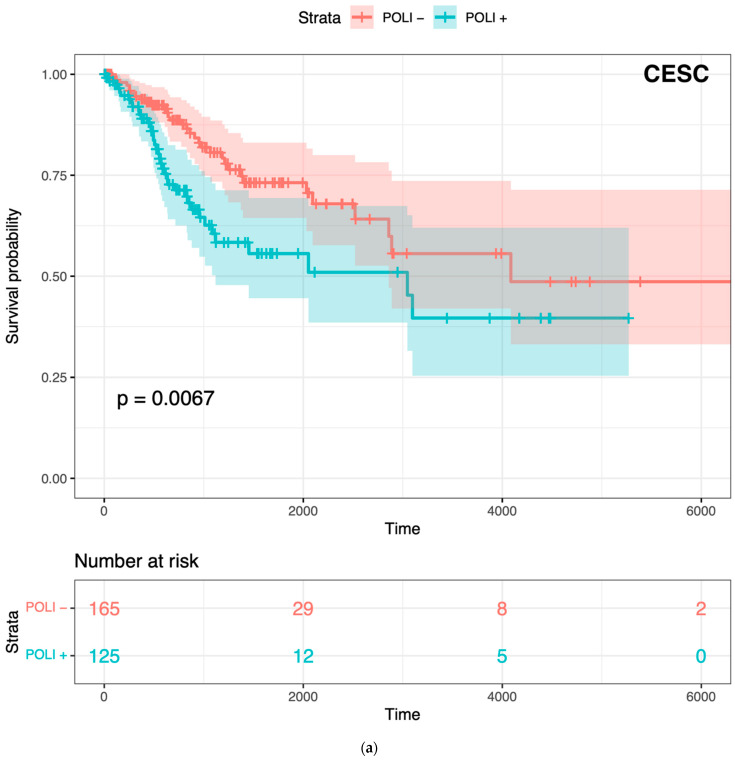
Prognostic value of *POLI* in CESC, PADD, and combined cancers: (**a**) Survival analysis of 291 patients in CESC vs. *POLI* expression; (**b**) Survival analysis of 176 patients in PAAD vs. *POLI* expression; (**c**) Survival analysis of 5213 patients in 11 cancer types.

**Figure 2 ijms-23-08571-f002:**
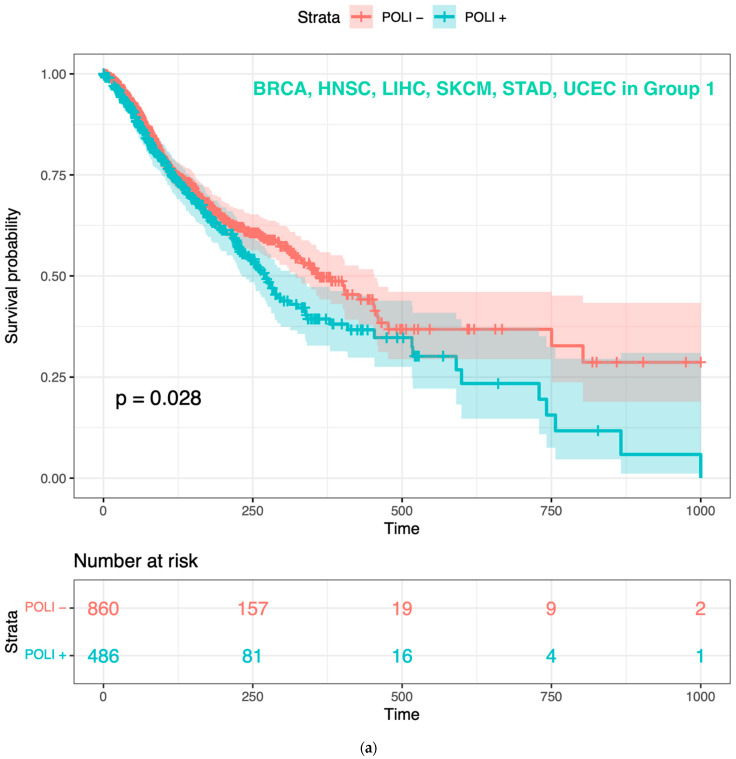
Reoccurrence of the prognostic value of *POLI* expression in regrouped cancer types: (**a**) K-M plot of *POLI* expression for patients in BRCA, HNSC, LIHC, SKCM, STAD, and UCEC from Group 1; (**b**) K–M plot of *POLI* expression for patients in BLCA, BRCA, COADREAD, LUAD, LUSC, OV, STAD, and UCEC from Group 2.

**Figure 3 ijms-23-08571-f003:**
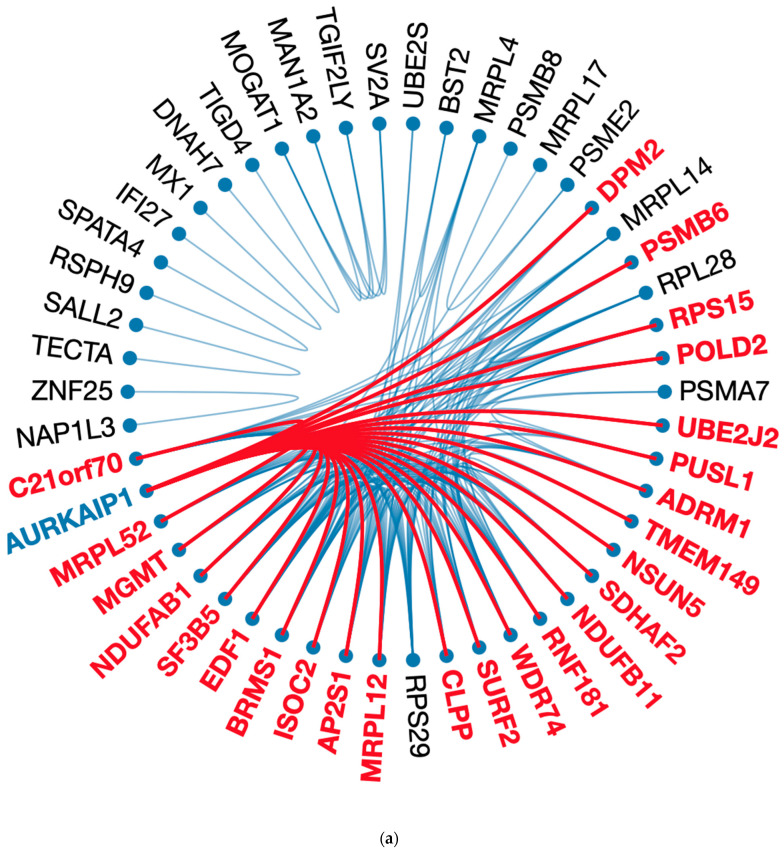
Visualization for most correlated genes regarding *POLI* expression in network and heatmap layouts: (**a**) Circosplot for 50 most *POLI*-related genes from 100 genes returned by RFC in CESC; (**b**) Circosplot for 50 most *POLI*-related genes from 100 genes returned by RFC in PAAD; (**c**) heatmap layout of 50 genes in (**a**); (**d**) heatmap layout of 50 genes in (**b**). For (**c**,**d**) on the *y*-axis locations, labels of the *POLI*-related genes were removed when they were not the top 50 important variables in RFC.

**Figure 4 ijms-23-08571-f004:**
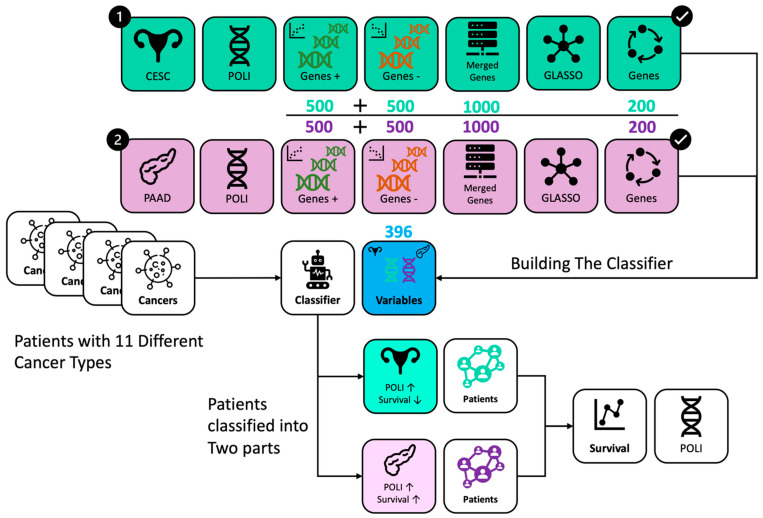
Workflow of data processing and analysis. Genes were selected from CESC in green and PAAD in purple, respectively. RFC incorporated 396 genes from CESC and PAAD. Patients with different cancer types were classified into two groups with the same *POLI* signature on survival as in CESC and PAAD.

**Table 1 ijms-23-08571-t001:** *POLI* vs. other gene correlation ranges in CESC and PAAD.

Caner	Top 500 Positive		Top 500 Negative	
	Max	Min	Min	Max
CESC	0.563 ^1^	0.311	−0.369	−0.225
PAAD	0.741	0.520	−0.577	−0.373

^1^ Each value indicates the Pearson correlation coefficient between *POLI* and another gene.

**Table 2 ijms-23-08571-t002:** The binary classification for 11 cancers based on the prognostic value of *POLI*-associated genes in CESC and PAAD.

Prediction	Cancer	Number
Group 1, Potentially *POLI* ↑ Survival ↓ ^1^	BLCA	291
	BRCA	483
	COADREAD	299
	HNSC	460
	LIHC	59
	LUAD	142
	LUSC	420
	OV	189
	SKCM	299
	STAD	258
	UCEC	134
Group 2, Potentially *POLI* ↑ Survival ↑ ^2^	BLCA	136
	BRCA	729
	COADREAD	134
	HNSC	106
	LIHC	364
	LUAD	434
	LUSC	132
	OV	118
	SKCM	174
	STAD	192
	UCEC	67

^1^ Group 1, when the result from RFC was CESC. ^2^ Group 2, when the result from RFC was PAAD. Arrows indicate that in group 1 when *POLI* expression is increased survival decreased and that in Group 2 when *POLI* expressing is increased survival increased.

**Table 3 ijms-23-08571-t003:** Gene components for enriched biological processes.

Biological Process	Gene ^1^	Rank in RFC ^2^
Mitochondrial translational elongation	*MRPL4* *MRPL14* *MRPL17* *MRPL12* *MRPL37* *MRPL52* *AURKAIP1*	5 6 8 27 63 86 92
Protein-containing complex subunit organization	*UBE2S* *MRPL4* *MRPL14* *MRPL17* *DNAH7* *MRPL12* *AP2S1* *SDHAF2* *NAP1L3* *IKZF4* *MRPL37* *NDUFAB1* *CENPF* *MX1* *RPS15* *KIF2C* *HMGA1* *MRPL52* *NDUFB11* *CELF4* *AURKAIP1* *SNAP25* *SV2A* *ADRM1*	3 5 6 8 24 27 29 38 52 56 63 71 72 79 81 84 85 86 87 90 92 95 97 99
Wnt signaling pathway, planar cell polarity pathway	*PSMB8* *PSMA7* *PSMB6* *AP2S1* *PSME2* *MAGI2*	1 2 10 29 31 77
Tumor necrosis factor-mediated signaling pathway	*PSMB8* *PSMA7* *PSMB6* *KRT8* *PSME2* *EDA*	1 2 10 23 31 91
Exocytic process	*CPLX2* *RAB3A* *SNAP25* *SV2A*	28 62 95 97

^1^ Color in the gene column. Gene in three different backgrounds means they were involved in 50 genes that were selected by GLASSO in CESC (green), PAAD (purple), or both (blue). ^2^ Rank in RFC, the importance of genes in RFC.

## Data Availability

All data analyzed in this report is publicly available. We include references to the original sources the data in the text.

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
