# Peer review of "Predicting the Prognostic Value of POLI Expression in Different Cancers via a Machine Learning Approach"

_ijms, 2022, doi:10.3390/ijms23158571_

Round 1
Reviewer 1 Report
This manuscript indicated novel prediction model of cancer survival using POLI expression by machine learning. This study is very important as new prognostic tool for different cancers. But, some corrections may be required. As correlation coefficient of POLI, Pearson correlation coefficient is used in Table 1. In addition, it is better to add other correlation coefficients such as spearman's rank correlation coefficient and Kendall rank correlation coefficient as non-parametric method. In addition, it was better to add other classifier algorithms than random forest, such as XGboost, LightGBM, neural networks, etc., and to compare these results. Also, it is better to add the advantages and disadvantages of this new strategy in discission section.
Line 287, page 10, typo:Forrest
Reviewer 2 Report
Comment 1.
In order for the authors to substantiate their scientific conclusion in the manuscript, the authors should check the following things in their datasets for the cancer patients.
First, how many of the cancer patients have mutations in the POLI gene ?
Second, is there a statistically significant difference in POLI mutation status between one group(POLI high expression showing worsening survival) and other group(POLI high expression showing improved survival) ? If so, their scientific conclusion in the manuscript should not be based on present mechanistic explanation, but otherwise mechanistic explanation.
